# Bodyweight change and cognitive performance in the older population

**Judith M. Kronschnabl**[1]*, **Thorsten Kneip**[1], **Luzia M. Weiss**[1], **Michael Bergmann**[1,2]

**1** Munich Center for the Economics of Aging (MEA), Max Planck Institute for Social Law and Social Policy, Munich, Germany, **2** Technical University of Munich (TUM), Chair for the Economics of Aging, Munich, Germany

* kronschnabl@mea.mpisoc.mpg.de

**Data Availability Statement:** The data underlying this study are owned by SHARE (http://www.share-project.org) and can be accessed following the protocol outlined in the Methods section (DOIs: 10.6103/SHARE.w2.600, 10.6103/SHARE.w4.600, 10.

## Abstract

Preservation of cognitive function is one of the major concerns in contemporary ageing societies. At the same time, overweight and obesity, which have been identified as risk factors for poor health development, have been increasing in many countries all over the world. This study examines the relationship between bodyweight change and cognitive decline in old age and it aims to determine whether and how changes in body mass index (BMI) affect the development of cognitive functioning in old age. Using longitudinal data from the Survey of Health, Ageing and Retirement in Europe (SHARE), covering four waves between 2006 and 2016 with 58,389 participants from 15 countries aged 50+, we estimated asymmetric fixed effects models by gender, adding possible confounding variables such as age, grip strength, health conditions, and physical activity. Additionally, we investigated possible heterogeneity in the BMI-cognition relation. We found a positive association between BMI change and change in cognitive performance, which was dominantly driven by BMI decrease. Weight loss was typically negatively related to cognition, particularly at low levels of BMI and mainly due to health conditions affecting both bodyweight and cognitive performance. Weight gain was, on average, not significantly related to cognitive performance; only respondents with preceding weight loss profited from small increases in BMI. Our analyses provide no support for an "obesity paradox" in cognition, according to which higher weight preserves cognition in old age. The association between weight change and cognitive performance in older age is based on weight changes being related to illness and recovery.

## Introduction

According to the World Health Organization (WHO), the prevalence of overweight and obesity in later life has increased dramatically throughout the world, a trend that poses serious challenges to public health and healthcare systems [1]. At the same time, cognitive decline is one of the major concerns of the older population, as preservation of cognitive function is essential for maintaining quality of life [2]. Understanding the relationship between late life weight and cognitive performance is therefore crucial for an ageing society. While the literature provides conclusive evidence that overweight and obesity are negatively associated with

6103/SHARE.w5.600, and 10.6103/SHARE.w6. 600).

**Funding:** None of the authors of the manuscript has received specific funding for the work included in this submission. The funders mentioned below did not play any role in the study design, data analysis, decision to publish, or preparation of the manuscript. The SHARE data collection has been primarily funded by the European Commission through FP5 (QLK6-CT-2001-00360), FP6 (SHARE-I3: RII-CT-2006-062193, COMPARE: CIT5-CT-2005- 028857, SHARELIFE: CIT4-CT-2006-028812) and FP7 (SHARE-PREP: N°211909, SHARE-LEAP: N°227822, SHARE M4: N°261982). Additional funding from the German Ministry of Education and Research, the Max Planck Society for the Advancement of Science, the U.S. National Institute on Aging (U01_AG09740-13S2, P01_AG005842, P01_AG08291, P30_AG12815, R21_AG025169, Y1- AG-4553-01, IAG_BSR06-11, OGHA_04-064, HHSN271201300071C) and from various national funding sources is gratefully acknowledged (see www.share-project.org).

**Competing interests:** The authors have declared that no competing interests exist.

cognitive performance in children and adults [3–6]; the link between bodyweight and cognition for the older population seems to be more complex and is not yet fully understood [7]. For this group, there is some evidence suggesting that overweight or even obesity may attenuate cognitive decline and optimal weight might be higher in old age than in midlife [8–12]. This phenomenon has sometimes been referred to as the "obesity paradox" in cognition [13–16].

For example, Kou and colleagues [15] found that overweight persons performed better on reasoning tests and visuospatial processing speed compared to their normal-weight counterparts in a cross-sectional sample of older adults 65+ in the United States. Similarly, Skinner et al. [17] report significantly better cognitive flexibility performance for both, younger (55+) and older (75 +) obese old persons, compared to those of normal weight. However, similar to the results found for children and adults, other studies have found a negative association between bodyweight and cognition also for older people [18–20], while some suggest that leanness rather than overweight is associated with poor cognitive [21]. This raises the question of how to aggregate these inconclusive findings, given that studies arguably differ in several respects like, for instance, features of the study sample. Based on a review of the literature, Smith et al. [7] have suggested that the mean age of the study sample may be a crucial factor with a positive weight-cognition association only emerging in samples with mean age above 72 years. However, the subsequent studies of neither Skinner et al. [17] nor Benito-Leon et al. [18] could confirm this: the former indicated a positive relationship also for the younger old, whereas the latter found a negative association in a relatively old sample with mean age about 77 years.

A limitation of most studies on the relationship between bodyweight and cognition in old age is that they are cross-sectional. Longitudinal studies would be generally preferable as they offer the possibility to draw causal conclusions under weaker and more plausible assumptions when appropriate methods are employed. Within the longitudinal literature, there is rather unanimous support from prospective studies that being overweight or obese in midlife is positively associated with poorer cognitive performance in late life [22–24]. However, when looking only at older persons and employing longitudinal methods, findings are again contradictory. A study by Memel et al. [25] found that in a large cross-national European sample of community-dwelling older adults, lower body mass predicted better cognition at baseline, but *changes* in body mass were positively associated with *changes* in cognition over time. In other words: although an elevated initial BMI seemed to be detrimental to cognitive performance, gaining weight in old age appeared to mitigate ageing-related cognitive decline. This study not only supports the obesity paradox in cognition but also interprets the concept in a longitudinal perspective: weight *gain* may be beneficial for maintaining cognitive functioning; weight *loss* (even in the obese), on the other hand, should be avoided [26–28]. However, the study has some shortcomings that limit the interpretation of findings. Although possible confounding diseases were discussed, they were not empirically taken into account; possible gender-specific effects were neglected and, maybe most importantly, the effect of weight change was modelled symmetrically. Thus, no conclusions could be drawn as to whether the observed positive association of bodyweight and cognition was owed to weight gain or weight loss, although previous research has suggested that weight change in either direction might be detrimental [29]. Driscoll and colleagues [28] analysed how transitions between BMI categories affected changes in cognitive performance. While such an approach accounts for possible asymmetric effects of weight change, it comes at the cost of not fully exploiting the available information, since weight changes within a category are neglected. They found no association between weight gain and cognitive performance in a large multi-ethnic community-dwelling sample of postmenopausal women, while weight loss was associated with worse cognitive performance. There might be different effects not only according to the direction of weight

change but also according to gender, as shown by Han et al. [30]. In a sample of older Koreans, they found that weight gain was associated with a positive change in cognitive function for men who were obese at baseline assessment. For women in turn, weight loss was associated with cognitive decline, independent of their baseline assessment. Yet, this study is limited by its small sample size and the relatively short follow up period of only two years.

Given the state of research, there is still a need to explain why some studies found a positive association of weight (change) and (change in) cognitive performance in the old while others do not. A detailed examination of the possible mechanisms that may lead to these conflicting findings is therefore indispensable.

## The obesity paradox in cognition: Possible explanations

To explain the obesity paradox in cognition, several physiological mechanisms have been discussed. On the one hand, obesity negatively affects the brain structure by reducing grey matter density [31,32]. Furthermore, an increased BMI promotes cerebral inflammation [16] and elevated levels of C-reactive protein [33], which may cause cognitive decline. Consequently, people who are overweight or obese in midlife are more likely to experience cognitive impairment in later life. On the other hand, adipose tissue produces leptin [34] and increases myelin through elevated white matter volumes [31,32], which in turn benefits cognition.

While biological pathways could go in either direction, alternative explanations point to several methodological issues. In this respect, part of the obesity paradox could be due to BMI, the most common measure for obesity used in the literature, being largely uninformative about body composition. Looking only at BMI, a decrease in skeletal muscle mass (sarcopenia), which is common in older adults, would be undiscerned if accompanied by an increase in fat mass [35]. To account for differences and changes in body composition, other measures, such as waist circumference or grip strength, might be better measures of fat or functional skeletal muscle mass, respectively [36]. When considered simultaneously or to complement BMI, they would allow for a differentiation between possible effects of fat vs. lean body mass. Furthermore, weight change and cognitive impairment may co-occur with other morbidities, particularly in older age. Underlying diseases, some of which may be unobserved or even unknown, might affect both bodyweight and cognitive function and thus introduce a spurious correlation. While the inclusion of available health-related control variables is common practice, there has been little reflection on their selection. Some, e.g., hypertension or cardiovascular diseases, might actually be intermediate mechanisms rather than confounding factors [37]. Adjusting for such factors would then eliminate potential negative effects of overweight on cognition via these factors, resulting in upward bias.

A serious objection raised against findings that seem to support an obesity paradox in cognition is the possibility of "survivor bias" in a sample of older individuals [7]: If overweight or obesity as well as poor cognitive functioning share common unobserved causes with mortality or, more generally, other sources of study non-participation, this will introduce a positive spurious association between bodyweight and cognition in the observed older population. This is because, conditional on "surviving", those exhibiting one risk factor (overweight) will more likely lack the other (poor cognition). Unfortunately, only few studies have attempted to address this issue. However, the problem should generally be less severe when longitudinal methods based on differencing are employed. Here, the corresponding problem to survival bias is attrition bias. Unlike with cross-sectional methods, attrition may be correlated with time-constant unobserved heterogeneity (e.g. genetic dispositions) without introducing bias. Moreover, it can be relatively easily assessed whether attrition bias is an actual concern in a present analysis [38].

Finally, when investigating effects of weight change (or intrapersonal variation in weight) in a longitudinal setting, it must be taken into account that changes in either direction might affect cognition in the same direction, but to different degrees. This possibility would be ruled out by design when regressing changes in cognition on changes in weight, since this imposes effects of equal size but in the opposite direction. In the presence of detrimental cognitive consequences of weight change, with weight loss being more harmful then weight gain, such an approach would produce a positive weight effect, which might easily lead to the false conclusion that weight gain is beneficial. In addition, effects of weight change on cognition could be different depending on the age or BMI level at which they occur. In other words, reducing bodyweight by the same amount could have different consequences for someone who is obese compared to someone who is already underweight. Moreover, weight gain could indicate a possible recovery effect, especially if bodyweight is regained after a disease-related weight loss. To shed light on the underlying mechanisms, statistical methods must therefore take into account all these possible nonlinearities and heterogeneous effects.

Overall, the literature provides some, yet inconclusive evidence on a positive relation between bodyweight and cognition in old age. The state of findings has given rise to various interpretations. Some are in favour of the existence of an obesity paradox and have contributed to the common perception that a little extra weight may not only do no harm but, in fact, be beneficial in old age. Others question the informative value of those findings against the background of the methodological issues involved. This study contributes to the discussion by carefully examining the relationship between bodyweight and cognitive performance in a large cross-national sample of older Europeans, making full use of the advantages offered by its panel structure and addressing all the aforementioned methodological concerns.

## Methods

### Data source

The following analyses use data from Waves two, four, five, and six of the Survey of Health, Ageing and Retirement in Europe (SHARE) [39,40] that can help mitigate most of the mentioned issues. The SHARE study is subject to continuous ethics review. During Waves 1 to 4, SHARE was reviewed and approved by the Ethics Committee of the University of Mannheim. Wave 4 of SHARE and the continuation of the project were reviewed and approved by the Ethics Council of the Max Planck Society. SHARE is a multidisciplinary panel study providing information on health, socioeconomic status, and social and family networks of respondents aged 50 and over. From 2004, data were collected every two years. By its sixth wave, SHARE included 20 European countries plus Israel. The sample was restricted to respondents repeatedly observed without any missing information on the variables used for analysis. Since parts of our analyses required at least three observations in time per respondent, countries that had participated in less than three panel waves were excluded for comparability. This left us with an analytic sample of 32,467 women (in 87,777 observations) and 25,922 men (in 69,203 observations) from 15 countries (Austria, Belgium, Czech Republic, Denmark, Estonia, France, Germany, Israel, Italy, Netherlands, Poland, Slovenia, Spain, Switzerland), observed in at least two waves.

### Measures

**Cognition.** As a measure of cognitive function, we created a standardized index based on immediate and delayed word recall tasks [41]. Based on a modified version of the Rey's Auditory Verbal Learning Test (RAVLT), respondents were asked twice to recall as many words as possible from a list of ten simple nouns within one minute: first, immediately after the words had been read out (memory) and again after some further interview questions (recall) [42].

We focused on these measures because fluid cognitive skills, like memory and recall, involving mastering new tasks and remembering and processing new information, have been found to be affected first and more pronounced by cognitive ageing. In contrast, crystallised cognitive skills, acquired and learned in the past, such as numerical and verbal skills, have been shown to remain relatively stable over the life course [43–47]. Moreover, scores obtained from the recall tests exhibit some desirable properties: they are approximately normally distributed, non-skewed, and do not suffer from floor or ceiling effects [45].

**Bodyweight.** We used BMI as measure for (height adjusted) bodyweight. The score was calculated as $BMI = weight/height^2$ using self-reported weight (in kg) and height (in m). While weight was asked in every wave to capture changes, height was only asked once and was assumed to remain stable over all observations. In the empirical part of the paper, we use the terms BMI and bodyweight interchangeably.

**Reasons for bodyweight change.** Respondents who reported weight loss were asked for its reason. This question was posed to all respondents in waves five and six who explicitly reported a weight loss during the last 12 months, while in waves two and four, the question was only asked to respondents for whom the reported bodyweight was at least five kg lower than in the preceding observation. Answer options were collapsed into two categories distinguishing between reasons related to and reasons unrelated to illness.

**Grip strength.** To capture changes in body composition, we used information on respondents' grip strength. Hand grip strength has been found to be significantly correlated with fat-free (lean) body mass in men [48] and women [49]. As a proxy for unobserved conditions associated with a detrimental reduction in lean body mass (sarcopenia), the grip strength measure–as a complement to BMI–allowed us to differentiate between weight loss attributable to a reduction in body fat and weight loss attributable to a reduction in muscle mass. For the models in this article, we used the maximum value (in kg) out of up to four measurements from both hands (two measurements per hand).

**Physical activity.** Physical activity, as a (mostly intentional) reason for weight loss or maintenance, was measured on a four-point scale, where respondents were asked to rate the frequency of vigorous physical activity, such as sports, heavy housework, or a job involving physical labour, as "hardly ever or never", "1–3 times a week", "once a week", or "more than once a week".

**Observed diseases.** We identified certain health conditions that might lead to (unintentional) weight changes as well as to cognitive decline and that are observed in all waves. These are Parkinson's disease (PD) [50], stroke [51], all types of cancer (except for minor skin cancers) and the associated chemotherapy [52,53], dementia [54,55] and depression [56]. We do not control for conditions, for which increased bodyweight is a known risk factor, like cardiovascular diseases. This would lead to "overcontrol" by eliminating a mechanism by which weight change may affect cognitive performance.

**Demographics.** We included a quadratic function of age in our models to account for the fact that both BMI and cognition change over time, while cognitive decline might accelerate with age. We chose a quadratic parametrization for reasons of parsimony; further checks revealed that a fully flexible modelling did neither improve model fit nor alter regression coefficients of interest. Cross-sectional models were additionally adjusted for educational qualification, measured by the International Standard Classification of Education (ISCED), and differences between countries by including country dummies. Furthermore, all models were calculated separately for male and female respondents.

Table 1 provides summary statistics on the employed measures in the analytic sample. The unstandardized score for immediate and delayed word recall ranged from 0 to 20 with mean 9.3 and standard deviation 3.6. BMI increase and decrease refer to average changes between two measurements.

**Table 1. Summary statistics.**

|  | Men | Women | Total |
|---|---|---|---|
| Cognition score | -0.10 | 0.08 | 0.00 |
| (Word recall) | (0.96) | (1.02) | (1.00) |
|  | [-2.62; 2.93] | [-2.62; 2.93] | [-2.62; 2.93] |
| BMI | 27.15 | 26.67 | 26.88 |
|  | (3.99) | (4.91) | (4.54) |
|  | [11.02; 77.03] | [12.80; 64.02] | [11.02; 77.03] |
| BMI increase | 0.63 | 0.76 | 0.70 |
|  | (1.19) | (1.38) | (1.30) |
|  | [0; 19.53] | [0; 18.88] | [0; 19.53] |
| BMI decrease | 0.64 | 0.71 | 0.68 |
|  | (1.28) | (1.44) | (1.37) |
|  | [0; 17.93] | [0; 29.42] | [0; 29.42] |
| Age | 67.05 | 66.85 | 66.94 |
|  | (9.32) | (9.69) | (9.52) |
|  | [50; 102.3] | [50; 103.8] | [50; 103.8] |
| Grip strength (kg) | 41.23 | 24.71 | 32.00 |
|  | (13.15) | (9.31) | (13.86) |
|  | [0; 98] | [0; 100] | 0; 100] |
| Cancer | 0.05 | 0.05 | 0.05 |
|  | (0.21) | (0.21) | (0.21) |
|  | [0; 1] | [0; 1] | [0; 1] |
| Parkinson's disease | 0.01 | 0.01 | 0.01 |
|  | (0.09) | (0.08) | (0.08) |
|  | [0; 1] | [0; 1] | [0; 1] |
| Stroke | 0.04 | 0.03 | 0.03 |
|  | (0.20) | (0.17) | (0.18) |
|  | [0; 1] | [0; 1] | [0; 1] |
| Dementia | 0.01 | 0.01 | 0.01 |
|  | (0.10) | (0.10) | (0.10) |
|  | [0; 1] | [0; 1] | [0; 1] |
| Depression | 0.18 | 0.32 | 0.26 |
|  | (0.39) | (0.47) | (0.44) |
|  | [0; 1] | [0; 1] | [0; 1] |
| Other disease | 0.15 | 0.17 | 0.16 |
|  | (0.36) | (0.38) | (0.37) |
|  | [0; 1] | [0; 1] | [0; 1] |
| *Physical activity* hardly ever | 0.37 | 0.46 | 0.42 |
|  | (0.48) | (0.50) | (0.49) |
|  | [0; 1] | [0; 1] | [0; 1] |
| 1–3 times per month | 0.09 | 0.08 | 0.09 |
|  | (0.29) | (0.28) | (0.28) |
|  | [0; 1] | [0; 1] | [0; 1] |
| Once a week | 0.14 | 0.14 | 0.14 |
|  | (0.35) | (0.35) | (0.35) |
|  | [0; 1] | [0; 1] | [0; 1] |
| More than once a week | 0.40 | 0.31 | 0.35 |
|  | (0.49) | (0.46) | (0.48) |

(*Continued*)

**Table 1.** (Continued)

|  | **Men** | **Women** | **Total** |
|---|---|---|---|
|  | [0; 1] | [0; 1] | [0; 1] |

Standard deviation in parentheses, range in brackets.

## Statistical analysis

Our analyses were structured sequentially, i.e. we employed a series of model specifications to challenge and relax the underlying assumptions required for causal inference. As a starting point, we took the association of bodyweight and cognitive performance in our sample as described by regression model

$$y_{it} = \alpha + \beta_{POLS}x_{it} + \alpha_i + \varepsilon_{it}, \tag{1}$$

where $y_{it}$ and $x_{it}$ represent our measures for cognitive performance and bodyweight, respectively, for any observation in the pooled sample. $\alpha_i$ and $\varepsilon_{it}$ denote the time-constant and time-varying components of the error term. In order to give $\beta_{POLS}$, the pooled OLS estimator, a causal interpretation, $x_{it}$ must not be correlated with either of the error components. Since this assumption is likely violated, further adjustment is necessary. The standard approach here is to control for observable characteristics (C), which are assumed to potentially confound the relationship of interest. As a first adjustment, we included age, education, and country dummies as observed controls.

$$y_{it} = \alpha + \beta_{POLS}x_{it} + \omega_{POLS}C_{it} + \alpha_i + \varepsilon_{it}, \tag{1A}$$

The next adjustment we made was splitting the association captured in a between and within component by regressing cognitive performance on the person-specific means of x and the deviations from the person specific mean (the "demeaned" variables):

$$y_{it} = \alpha + \beta_{BE}(\bar{x}_i) + \beta_{FE}(x_{it}-\bar{x}_i) + \omega_{BE}(\bar{C}_i) + \omega_{FE}(C_{it}-\bar{C}_i) + \alpha_i + \varepsilon_{it}, \tag{2}$$

where $\beta_{FE}$ gives the fixed effects estimator, which can also be obtained by estimating a regression on the demeaned data alone:

$$\tilde{y}_{it} = \beta_{FE}\tilde{x}_{it} + \omega_{FE}\tilde{C}_{it} + \tilde{\varepsilon}_{it}, \tag{3}$$

with $\tilde{y}_{it} = (y_{it}-\bar{y}_i)$, $\tilde{x}_{it} = (x_{it}-\bar{x}_i)$ and $\tilde{C}_{it} = (C_{it}-\bar{C}_i)$, such that all time-constant elements in C are cancelled out, which in our case reduces C to age. The fixed effects estimator has a more natural interpretation as how a *change* in bodyweight might affect the *change* in cognitive performance, since it does not rely on a between-person comparison. Moreover, and more importantly, it allows for an arbitrary correlation of $x_{it}$ and $\alpha_i$ without introducing bias. This also makes further adjustment on time-constant covariates–like genetic dispositions, but also largely stable socioeconomic and sociocultural conditions–unnecessary. In contrast, $\beta_{BE}$ is purely based on a comparison of respondents of different (average) weight. Its interpretation as effect of *being* of higher weight is thus threatened by omitted variable bias and would require further adjustment.

For the remaining parts of the analysis, we focused on intrapersonal weight change and extensions of the fixed effects approach. A shortcoming of the standard FE model is that it forces the estimated relationship to be symmetric: it assumes that an increase in BMI promotes cognitive performance by the same amount as a decrease reduces it, which may not be

warranted. As recently shown by Allison (2019) [57], this assumption can be easily relaxed by estimating

$$\tilde{y}_{it} = \beta^+ \tilde{z}^+_{it} + \beta^- \tilde{z}^-_{it} + \omega a\tilde{g}e_{it} + \tilde{\varepsilon}_{it}, \tag{4}$$

where ~ denotes the within transformation as in (3) and

$$z^+_{it} = \sum_{s=1}^{t} x^+_{is}; \ z^-_{it} = \sum_{s=1}^{t} x^-_{is}, \tag{5}$$

where $x^+_{it} = x_{it} - x_{it-1}$ if $(x_{it} - x_{it-1}) > 0$, otherwise 0 and $x^-_{it} = -(x_{it} - x_{it-1})$ if $(x_{it} - x_{it-1}) < 0$, otherwise 0. That is, instead of BMI, weight gain and weight loss are accumulated up to time t and used as regressors in a fixed effects regression. Note that $\beta_{FE}$ is a weighted average of $\beta^+$ and $-\beta^-$. This implies the possibility of obtaining $\beta_{FE} > 0$, suggesting that weight gain positively impacts cognitive performance, even if weight gain actually had detrimental cognitive effects ($\beta^+ < 0$)–namely if any weight change was bad but weight loss was worse than weight gain. In the results section, we refer to model (5) as asymmetric fixed effects model (FE-a).

At this stage, we added grip strength, the observed diseases described above, and physical activity as possible time-varying confounders. Note that the asymmetric specification of the model also allows these variables to confound the association of weight loss and cognition differently than the association of weight gain and cognition. To account for unobserved diseases that might be responsible for changes in bodyweight, we also ran a modified version of the model including the self-reported reason for weight loss. As described in the data section, this information was only available for a subsample, but for those individuals, it offered a straightforward approach to distinguish (unintentional) weight loss due to illnesses from that due to other reasons (e.g., physical activity or caloric restrictions). This resulted in model specifications

$$\tilde{y}_{it} = \beta^+ \tilde{z}^+_{it} + \beta^- \tilde{z}^-_{it} + \omega a\tilde{g}e_{it} + \gamma \tilde{W}_{it} + \tilde{\varepsilon}_{it} \text{ and} \tag{6}$$

$$\tilde{y}_{it} = \beta^+ \tilde{z}^+_{it} + \beta^- \tilde{z}^-_{it} + \beta^{ill} \tilde{z}^{ill}_{it} + \omega a\tilde{g}e_{it} + \gamma \tilde{W}_{it} + \tilde{\varepsilon}_{it}, \tag{7}$$

respectively, where $W_{it}$ is the vector of covariates and $z^{ill}_{it}$ is the accumulated illness-related weight loss up to time t, as reported by respondents. In the results section, we refer to models (6) and (7) as FE-a+ and FE-aII+, respectively.

A comparison of estimates from models with and without including $\tilde{W}_{it}$ yielded the total bias from omitting all considered control variables. In order to assess the relative contribution of grip strength, observed diseases and physical activity we decomposed the total bias by computing $v^{*'}\gamma^*$, where $\gamma^*$ is a vector of regression coefficients from (6) involving the elements in W (related to age, grip strength, diseases, and physical activity) and $v^*$ is a vector of regression coefficients from a set of auxiliary regressions

$$\tilde{w}_{it} = v^+ \tilde{z}^+_{it} + v^- \tilde{z}^-_{it} + \tilde{\zeta}_{it}, \tag{8}$$

on these elements (for details and inference see [58]).

A crucial assumption for fixed effects regression to yield consistent estimates of the causal effects is that age profiles of cognitive performance would be parallel for respondents absent weight change. This assumption would be violated in the presence of pre-existing trends in cognitive performance, which are related to subsequent weight change. For example, genetic or metabolic differences or health behaviour in childhood and early adulthood might be linked to both, onset and speed of cognitive decline and the propensity to maintain a stable weight. In fact, there is recent evidence that genetic dispositions, possibly in combination with socio-

economic factors, influence rates of decline in cognition and particularly episodic memory [59,60]. Accordingly, a final specification addressed this issue by allowing for individual age trends by "de-trending" the data, resulting in the fixed effects individual slopes (FEIS) estimator [38]: 377–81; [61].

$$\ddot{y}_{it} = \beta_{FEIS}^{+}\ddot{z}_{it}^{+} + \beta_{FEIS}^{-}\ddot{z}_{it}^{-} + \omega\,a\ddot{g}e_{it} + \gamma_{FEIS}\ddot{W}_{it} + \ddot{\xi}_{it}, \tag{9}$$

with $\ddot{y}_{it} = (y_{it} - \hat{y}_{it})$, $\ddot{z}_{it}^{+} = (z_{it}^{+} - \hat{z}_{it}^{+})$, etc. and $\hat{y}_{it}$ ($\hat{z}_{it}^{+}, \ldots$) denoting the predicted values of $y_{it}$ ($z_{it}^{+}, \ldots$) from their unit-specific fixed effects regressions on age. While assumptions for FEIS are weaker, it requires at least three observations in time per respondent, leading to a drop in sample size and efficiency.

As our analyses required at least two observations per respondent, we re-estimated all models employing inverse probability weighting to address potentially systematic dropout after baseline. Weights were obtained from logistic regressions of being observed only once on a set of baseline characteristics (BMI, health-related limitations, education, cohabitation status, area of living, country and age) in the full sample.

All analyses were carried out in Stata 14.2 [62], including additional packages st0085 [63], st0236 [58], and gr0059 [64]. We employed the user-written programs *xtfeis* and *xtart* to estimate fixed effects individual slope models and conduct specification tests for violation of the parallel trends assumption [61,65].

## Results

Fig 1 depicts the main results obtained from the different model specifications described in the previous section graphically. We provide detailed information on the underlying regression models in Table A (unweighted models) and Table B (weighted models) in S1 Table. Confidence intervals refer to the 95 per cent level and are based on cluster-robust standard errors.

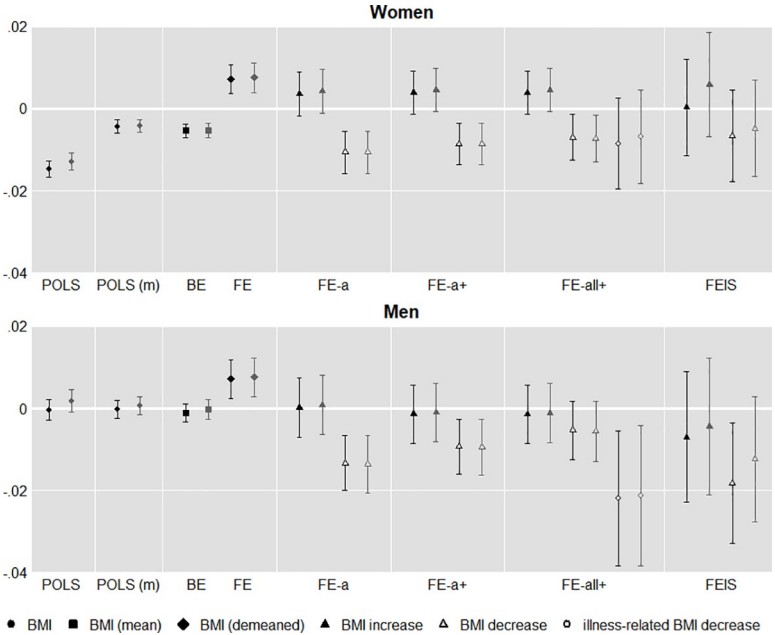

**Fig 1. Associations of BMI and cognitive performance estimated from different models.**

For all specifications, coefficients depicted in grey refer to respective estimates obtained from weighted models to account for potential initial panel dropout.

## Decomposing the association of BMI and cognitive performance

The overall association between BMI and cognitive performance in our pooled sample (POLS) was negative for women and not significantly different from zero for men. Controlling for age, education, and country (POLS (m), estimated by (1b)) considerably reduced the association for women, while it remained significant. Splitting the association into between- (BE) and within-individual (FE) components revealed that the overall association between BMI and cognitive performance was dominantly driven by the between association, while the within association was significantly positive for both men and women. These findings are largely in line with previous research. However, it is worth mentioning that, albeit significant, BMI explained very little of the variation (both, between and within) in cognitive performance and the effect sizes were very small: a one-unit increase in BMI shifted predicted cognition by 0.007 standard deviations, which translates to about 0.03 additional words in the combined immediate and delayed recall task.

Starting from these mere descriptive findings and focusing on changes in bodyweight, we investigated whether the positive association was produced by weight loss or weight gain. The two sets of estimates in the asymmetric fixed effects model (FE-a) in Fig 1 refer to the coefficients for weight gain (solid triangle) and weight loss (hollow triangle), estimated from Eq (4), i.e. before including further control variables. Without further adjustment beyond age, we found no significant positive effect for weight gain neither for men nor for women, irrespective of whether we applied weights or not. However, there was a significant negative impact of weight loss.

## Addressing possible confounding

Adding grip strength, reported diseases, and physical activity as presumed time-varying confounders further reduced the effects of weight loss (FE-a+, estimated from Eq (6)). Noteworthy, when only considering BMI change ignoring its direction (i.e. estimating Eq (3) with controls) and pooling men and women, as e.g. Memel et al. (2016) [25] have done using the same data, we could replicate their finding of a significantly positive BMI effect (not shown). However, this does not imply a beneficial effect of weight gain but was solely due to the negative effect of weight loss.

Our data allowed us to conduct an additional test to assess whether the significant coefficient for BMI decrease points to a substantive effect or rather to unaccounted health issues. To this end, we utilized self-reported information on whether an experienced weight loss was illness-related and added a separate variable accounting for this (see Eq (7)). Results are shown in Fig 1 (FE-aII+), where the hollow circle represents the additional indicator. Note that this slightly changed the interpretation of the original variable to weight loss not attributed to illness by respondents. For men, we could observe that the coefficient for the original variable was, as expected, further reduced and became insignificant while the effect of the additional variable was much larger. This pointed to unobserved heterogeneity in illness even after controlling for observed diseases and some additional value in the self-reported measure to uncover it. However, the two coefficients did not differ significantly due to their large standard errors. Moreover, we did not find the same pattern for women.

A final model specification addressed the possibility that age profiles in weight and cognitive performance may be spuriously correlated. This is not sufficiently addressed by controlling for age and individual intercepts and thus leads to potentially biased results from fixed

effects regression [61]. Results from employing Eq (9) are reported in the final column in Fig 1 (FEIS). While point estimates differed from FE-a+, a specification test revealed that these differences are not significant, making FE-a+ the model of choice. Moreover, differences were less pronounced when employing weights, suggesting that the FEIS sample, which required at least three observations per respondent, got more selective.

## Investigating the role of confounders

Overall, our findings suggested that the association between BMI and cognition was amplified by confounding variables, such as severe diseases, leading to weight loss or changes in body composition *and* cognitive impairment, whereas there was no protective effect of weight gain in older age against cognitive decline. To learn more about the underlying mechanisms, we investigated how the different control variables contributed to the confounding effect. Table 2 reiterates the estimates for BMI increase and decrease before (FE-a) and after (FE-a+) including controls and decomposes the differences (Δ) according to the contributions of the used control variable sets.

For both, women and men, grip strength and observed diseases accounted for a sizeable and significant part of the association of weight loss and cognitive decline–over and above age, which might have already partially absorbed any (unobserved) age-related health deterioration, and physical activity. For men, the overall contribution amounted to about a third of the total within association. For women, the fraction was smaller (about one fifth). This was mainly due to the greater importance of grip strength in shaping the association of weight loss and cognitive decline for men, which might point to hand grip strength being a better indicator of lean muscle mass for men than for women [48]. Observed diseases confounded BMI decrease negatively for both, men and women; physical activity, although itself beneficial for cognitive performance, was not found to significantly confound the relationship.

The estimate for BMI increase, in turn, was positively confounded by grip strength, again more strongly for men. This clearly pointed to BMI alone being a suboptimal measure for

**Table 2. Disentangling the confoundedness of BMI change and cognitive performance.**

| women | | | | | | |
|---|---|---|---|---|---|---|
| | **β (s.e.)** | | **Δ** | **Δ attributable to** | | |
| | *FE-a* | *FE-a+* | *overall* | *grip strength* | *diseases* | *physical activity* |
| BMI increase | 0.0036 | 0.0039 | -0.0003 | 0.0003* | -0.0004* | -0.0003** |
| | (0.0027) | (0.0027) | (0.0003) | (0.0001) | (0.0002) | (0.0001) |
| BMI decrease | -0.0106*** | -0.0086*** | -0.0020*** | -0.0009*** | -0.0011*** | -0.0000 |
| | (0.0026) | (0.0026) | (0.0003) | (0.0002) | (0.0002) | (0.0001) |
| men | | | | | | |
| | **β (s.e.)** | | **Δ** | **Δ attributable to** | | |
| | *FE-a* | *FE-a+* | *overall* | *grip strength* | *diseases* | *physical activity* |
| BMI increase | 0.0003 | -0.0014 | 0.0016*** | 0.0016*** | 0.0003 | -0.0003* |
| | (0.0037) | (0.0036) | (0.0004) | (0.0003) | (0.0002) | (0.0001) |
| BMI decrease | -0.0133*** | -0.0092** | -0.0041*** | -0.0026*** | -0.0014*** | -0.0001 |
| | (0.0034) | (0.0034) | (0.0005) | (0.0004) | (0.0002) | (0.0001) |

Note

* $p<0.05$

** $p<0.01$

*** $p<0.001$; clustered SE in parentheses.

overweight as it largely ignores body composition. As grip strength itself was positively related to cognition in the within regressions on cognition, holding BMI constant, our findings can be interpreted as a positive effect of lean body mass at a given weight. Interestingly, while BMI increase was not significantly confounded by observed diseases for men, we found that it was negatively confounded for women (see Table 2). Illness thus appeared to be positively related to BMI changes in either direction for women. This suggests that both estimates, for BMI decrease and increase, would likely be downward biased in the presence of further unobserved heterogeneity in confounding health conditions for women.

**Is weight gain beneficial for the oldest old?.** The analyses reported so far did not consider any potential heterogeneity in effects of weight change. However, interaction effects of weight change with other variables may exist, such that our finding of no effect for weight gain and a negative effect for weight loss may be wrong for subgroups of our sample. For instance, as suggested by the literature, a positive weight effect might only emerge at older ages [7]. We therefore split the sample according to respondents' age when they entered the panel and replicated all analyses on the two resulting subsamples. The chosen cut-point was 65 years, resulting in a mean age of about 76 years for the older and 60 years for the younger subsample.

Results can be summarized as follows: First, in line with previous findings, estimates from the models exploiting between-variation (POLS and BE) were significantly higher in the older compared to the younger subsample. The negative association of BMI and cognition found for women in the full sample could not be replicated for the older sample. However, estimates did not turn positive. Second, estimates from models exploiting within-variation only, including FEIS, were more pronounced in the older than in the younger subsample, but differences were not significant. Further details are reported in the supplementary material (see tables C, D, and E in S1 Table). Third, with respect to the confounding effects of grip strength and illness, patterns largely prevailed but were again more pronounced among the older, except for physical activity, which appeared to be more important for the younger (see table F in S1 Table).

## The potential role of recovery

We further investigated whether weight gain might exhibit different effects for people with a preceding episode of weight loss. Given that weight loss typically indicates illness, subsequent weight gain might point to recovery and *therefore* be positively related to cognitive performance. To test this hypothesis, we interacted BMI increase with a lagged measurement of BMI decrease and added this term to our reference model (FE-a+), using different specifications. We found a positive interaction for women: weight gain without preceding weight loss had no effect on cognitive performance but became beneficial at relatively high levels of preceding weight loss. While the linear interaction was significant at the one per cent level, this finding was also robust against the specification of the functional form of the interaction. Fig 2 shows the resulting average marginal effects of a one-unit BMI increase conditional on the magnitude of a preceding weight loss for women and men, using a squared parametrization of lagged BMI decrease. Our findings suggested that the positive (non-significant) effect previously found for women was likely pointing to a recovery effect. For men, there was no significant interaction.

A replication of the analysis, conducted separately for the younger and older subsamples, showed that the effect found in the overall sample was more pronounced among older women, while it did not occur among the younger ones (see figure A in S1 Fig). This can be seen as consistent with the notion of a recovery effect, since disease-related weight loss (as a prerequisite for recovery-related weight gain) becomes more likely with older age.

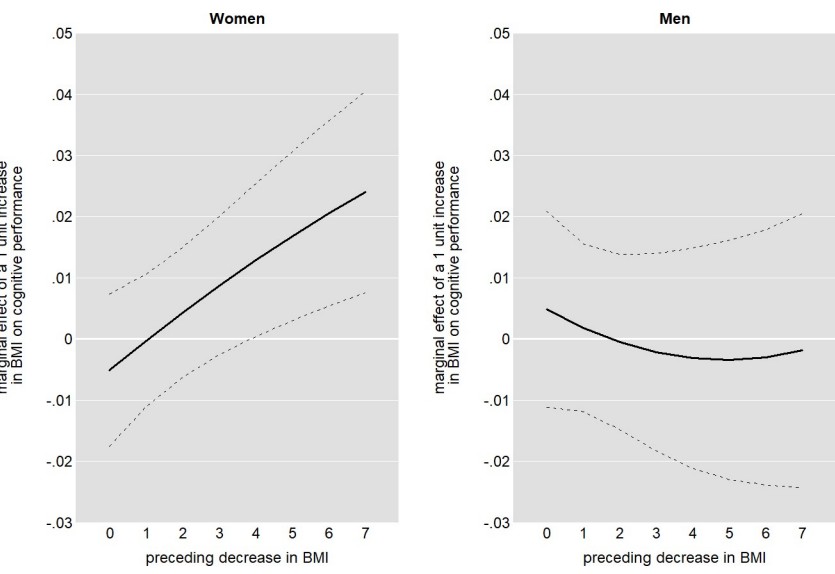

**Fig 2. Effects of BMI increase by preceding weight loss.**

## Further effect heterogeneity

Further effect heterogeneity could be at play: notably, the literature suggests positive effects of weight gain at higher levels of BMI and older age [7,30], the latter of which we already partially addressed. In addition, given the cross-national nature of our data and the mixed findings from different countries, accounting for possible heterogeneity across countries seemed appropriate. We addressed these issues by re-estimating Eq (6) including quadratic terms for the BMI change variables to account for non-linear effects, c.f. [29], as well as interactions with BMI level, age (squared) and country. Including a flexible specification of age in this exercise aimed for a more comprehensive understanding of possible age-heterogeneity, since any sample split is ultimately arbitrary. Based on this specification, we computed conditional average marginal effects as reported in Fig 3. Incremental F-statistics of the respective interaction terms were used to assess the presence of effect heterogeneity with respect to a specific variable.

We found non-linear effects for BMI increase as well as effect heterogeneity by BMI level and, to some extent, age (for women). The effect for BMI increase got smaller and eventually turned negative, the more weight people continued to accumulate. Similarly, while a BMI increase might be beneficial at low levels of BMI, the beneficial effect diminished and eventually turned negative at higher levels of BMI. For men, the reverse was true for weight loss, which particularly harmed the underweight but had no effect for the overweight. This might point to negative effects of weight loss (indicating unobserved illness) and actual positive effects cancelling each other out. Age patterns appeared less clear but provided no support for a positive effect of weight gain particularly for the oldest old. However, weight loss became more detrimental with women's age, which might lead to wrong conclusions if weight gain and loss are not considered separately.

We further found large country heterogeneity in our sample but no clear pattern (see figure B in S1 Fig). By tendency, effects were below average for the Baltic countries and above average for Central European countries and Israel. Interestingly, in countries where weight loss was more detrimental, weight gain was also more detrimental and vice versa. Countries thus appeared to differ in the direction and intensity by which older people respond cognitively to weight change in either direction.

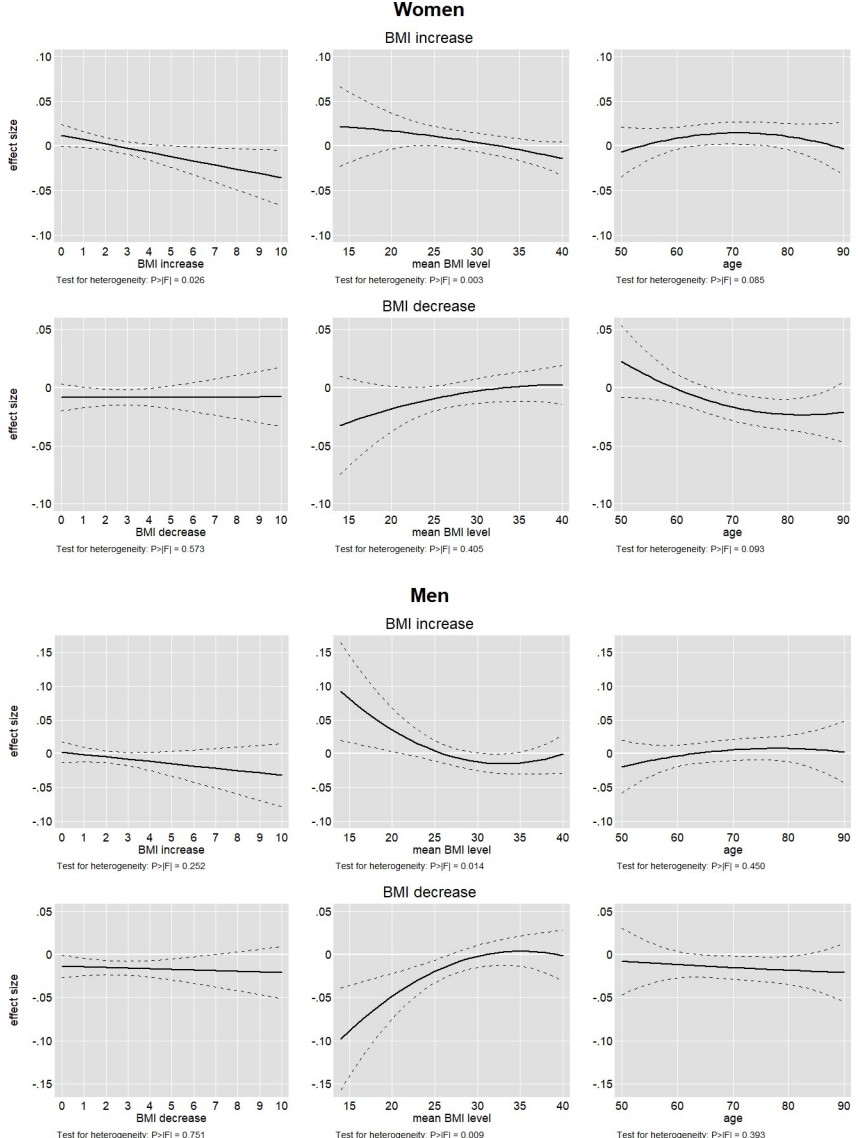

**Fig 3. Non-linearity and moderation by BMI level and age.**

## Sample selection

Our results rely on the analysis of repeated observations, which raises the question of selective panel dropout. To check whether sample selection drove our findings, we conducted two tests, one related to possible selectivity in entering the panel (i.e. being observed at least twice) and one related to panel retention. First, we applied inverse probability weighting to all models. Effectively, this put a higher weight on respondents in the panel sample that were more similar to initial dropouts in their baseline characteristics. As already shown in the results section, there were only marginal differences. Second, for all model specifications, weighted and unweighted, we implemented a formal test for attrition bias, which is based on the idea that a significant effect for a variable indicating dropout in the next wave points to a selective sample with respect to cognition [38]. The test was never significant for women or men in the full sample as well as in the younger subsample, reassuring that our estimates were not severely

affected by attrition bias. However, there was some indication of potential attrition bias in the models that did not account for confounding health issues in the older male subsample. This can be seen as in line with the notion of selective survival.

## Conclusion & discussion

This study aimed at examining the relationship between bodyweight change and cognitive decline in old age in a comprehensive way and, more specifically, challenge the hypothesis that weight gain may mitigate cognitive decline (the so-called obesity paradox in cognition). To this end, we considered asymmetric effects of weight change and a rich set of possible confounders, accounted for unobserved heterogeneity in cognition levels and age-trends and investigated the presence of heterogeneous effects across subgroups. Our findings demonstrate that the estimated (average) relationship crucially depends on whether the employed model imposes symmetric effects of weight gain and weight loss, a thorough consideration of confounding factors, and characteristics of the sample under study. This may well account for differences in previous findings.

Substantively, our findings suggest that weight loss in older age appears to be detrimental to cognitive development, but largely so because it is a sign of progressive physical deterioration. The remaining (i.e. conditional on observed illness) effect we found might be due to the incomplete or imperfect measurement of confounding factors. For example, there may be unobserved diseases leading to weight loss and cognitive decline at the same time. However, the remaining effect was also mainly driven by low-BMI respondents, for whom further weight loss may become a health issue itself. Our results further suggest that weight gain per se is, on average, unrelated to cognitive performance. Instead, a gain in bodyweight is disadvantageous when signalling illness or reduced physical activity, while it is beneficial when pointing to health recovery. However, we found some evidence for heterogeneous effects of BMI: while (particularly male) underweight persons might profit from a moderate weight gain, it becomes more and more detrimental as it accumulates or affects persons with an already elevated BMI. It should be stressed, though, that all found associations are substantially small and explain very little of the intrapersonal variability in cognitive functioning.

Of course, our study does not come without limitations. The first concerns the use of BMI, which does not provide information about body composition. While we addressed this limitation by complementing BMI with grip strength as a proxy for fat-free (lean) body mass, it must be noted that grip strength, although giving an approximation of total body muscle strength, is not a perfect measure. E.g., it is also dependent on the efficacy of the central and peripheral nervous systems to activate the muscles as well as participants' motivation or stamina [66]. More precise measures of body composition would be preferable, e.g. by means of bioelectric impedance analysis (BIA) or dual energy x ray absorptiometry (DXA) [67,68], which are, however, rarely available in large scale surveys. Alternatively, and more widespread, waist circumference or waist-to-hip ratio could be employed as a complementary measure of abdominal fatness [68]. Unfortunately, neither of these measures were available in the SHARE data.

A second limitation concerns the generalizability of our findings beyond possible weight effects on fluid cognitive ability as measured by immediate and delayed cognitive recall tests [43,69,70]. While this measure is affected first and most prominently by ageing [45], some previous (mostly cross-sectional) studies suggest that the link between weight and cognitive functioning may depend on the specific measure employed (see [7] for an overview). We thus encourage further research to see whether the findings from this study can be replicated using more refined measures for body composition and a more comprehensive view on cognitive functioning as multidimensional construct.

Overall, subject to the aforementioned limitations, our analyses thus provided no support for an obesity paradox in cognition. At the same time, they have some implications for the critical assessment and interpretation of previous findings. Treating weight change as a unidimensional concept presupposes that weight gain and weight loss have opposing effects, which may not be warranted. Studies focussing on BMI change may thus entice recipients to come to wrong conclusions about possible (positive) effects of weight gain, while, in fact, they are uninformative in this respect. The refutation of possible positive weight (gain) effects on cognition in old age has important practical implications. Given the known adverse health effects of being overweight and obese and the associated societal costs, physicians' false beliefs could prevent adequate medical advice. This may already be reflected in the documented rareness of doctoral advice to lose weight despite its effectiveness on behaviour [71].

The presence of effect heterogeneity we found implies that results will always be contingent on characteristics of the sample under study. For example, it will be more likely to find a positive association between weight gain and cognitive functioning the lower initial weight and the smaller the BMI increase experienced are on average and the more likely weight gain is an indication of health recovery. Additionally, there appears to be considerable cross-country variation in the way weight change and cognitive performance are related in the older population. This likely points to omitted relevant variables that systematically differ between countries and calls for more cross-cultural research on the topic.

## Supporting information

**S1 Fig. Heterogeneity.**
(PDF)

**S1 Table. Regression output.**
(PDF)

## Author Contributions

**Conceptualization:** Judith M. Kronschnabl, Thorsten Kneip, Luzia M. Weiss, Michael Bergmann.

**Data curation:** Thorsten Kneip, Luzia M. Weiss, Michael Bergmann.

**Formal analysis:** Thorsten Kneip.

**Methodology:** Judith M. Kronschnabl, Thorsten Kneip, Luzia M. Weiss, Michael Bergmann.

**Project administration:** Judith M. Kronschnabl.

**Validation:** Thorsten Kneip, Michael Bergmann.

**Visualization:** Thorsten Kneip.

**Writing – original draft:** Judith M. Kronschnabl, Thorsten Kneip, Luzia M. Weiss, Michael Bergmann.

**Writing – review & editing:** Judith M. Kronschnabl, Thorsten Kneip, Luzia M. Weiss, Michael Bergmann.

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
