## [Decision Letter · Decision Letter 0]

22 Jan 2021

PONE-D-20-38422

Bodyweight change and cognitive performance in the older population

PLOS ONE

Dear Dr. Kronschnabl,

Thank you for submitting your manuscript to PLOS ONE. After careful consideration, we feel that it has merit but does not fully meet PLOS ONE’s publication criteria as it currently stands. Therefore, we invite you to submit a revised version of the manuscript that addresses the points raised during the review process.

We look forward to receiving your revised manuscript.

Kind regards,

David Meyre

Academic Editor

PLOS ONE

Journal Requirements:

2.) Please ensure that you include a title page within your main document. We do appreciate that you have a title page document uploaded as a separate file, however, as per our author guidelines (http://journals.plos.org/plosone/s/submission-guidelines#loc-title-page) we do require this to be part of the manuscript file itself and not uploaded separately.

3.) Thank you for stating the following in the Funding Section of your manuscript:

'The SHARE data collection has been primarily funded by the European Commission

through FP5 (QLK6-CT-2001-00360), FP6 (SHARE-I3: RII-CT-2006-062193, COMPARE:

CIT5-CT-2005-028857, SHARELIFE: CIT4-CT-2006-028812) and FP7 (SHARE-PREP:

N°211909, SHARE-LEAP: N°227822, SHARE M4: N°261982). Additional funding from the

German Ministry of Education and Research, the Max Planck Society for the Advancement of

Science, the U.S. National Institute on Aging (U01_AG09740-13S2, P01_AG005842,

P01_AG08291, P30_AG12815, R21_AG025169, Y1-AG-4553-01, IAG_BSR06-11,

OGHA_04-064, HHSN271201300071C) and from various national funding sources is

gratefully acknowledged (see www.share-project.org).'

'The authors received no specific funding for this work.'

4.) We note that you have indicated that data from this study are available upon request. PLOS only allows data to be available upon request if there are legal or ethical restrictions on sharing data publicly. For information on unacceptable data access restrictions, please see http://journals.plos.org/plosone/s/data-availability#loc-unacceptable-data-access-restrictions.

5.) Please include captions for your Supporting Information files at the end of your manuscript, and update any in-text citations to match accordingly. Please see our Supporting Information guidelines for more information: http://journals.plos.org/plosone/s/supporting-information.

Reviewers' comments:

Reviewer's Responses to Questions

**Comments to the Author**

1. Is the manuscript technically sound, and do the data support the conclusions?

Reviewer #1: Yes

Reviewer #2: Yes

2. Has the statistical analysis been performed appropriately and rigorously? 

Reviewer #1: Yes

Reviewer #2: Yes

3. Have the authors made all data underlying the findings in their manuscript fully available?

Reviewer #1: No

Reviewer #2: No

4. Is the manuscript presented in an intelligible fashion and written in standard English?

Reviewer #1: Yes

Reviewer #2: Yes

5. Review Comments to the Author

Reviewer #1: This is an interesting paper examining the association between body weight change and cognitive performance. The methodology is impressive with good sample size. The analysis is careful and I agree that the paper is promising and may contribute to the literature.

Comments:

1. When discussing about obesity paradox, it is important to acknowledge and account for the limitation of using BMI as the only measure of obesity. Previous studies have showed that waist circumference and waist-to-hip predicted cognitive performance better than BMI. I understand that the authors used grip strength, but I am not clear how this can address the limitation of the lack of waist circumference measure. More elaboration is necessary and this limitation should be noted in the discussion.

2. Another limitation that should be acknowledged and discussed is the cognitive measures employed in the current study. Only a short immediate and delayed recall tasks were used. The authors should discussed the validity of these tasks in measuring cognitive functions. How reliable is the tasks? Some previous studies have found that the link between obesity cognitive functions may depend on the tasks and the specific cognitive abilities that were measured. More discussion is necessary and possible limitations should be acknowledged appropriately.

Reviewer #2: The manuscript shows an interesting human study, which could have potentially significant implications in the field of understanding the relationship between BMI and cognition. The study as presented shows that, weight gain is, on average, not significantly related to cognitive performance, concluding that the association between weight change and cognitive performance in older age is based on weight changes being related to illness and recovery.

However, there are some concerns related with this work:

Major revision:

The selected participant average age is 66,94 years. As commented by the authors based on a review of the literature, Smith et al. have suggested that the mean age of the study sample may be a crucial factor with a positive weight-cognition association only emerging in samples with mean age above 72 years. However the chosen age to perform the study is an age that is closer from the adulthood rather than elderly. Therefore, the study should be completed adding another group above 72 years, as suggested by the literature.

Minor revision:

Figure legends are missing.

6. PLOS authors have the option to publish the peer review history of their article (what does this mean?). If published, this will include your full peer review and any attached files.

Reviewer #1: No

Reviewer #2: No

---

## [Author Response · Author response to Decision Letter 0]

19 Mar 2021

Dear reviewers,

thank you very much for your valuable comments and the opportunity to revise our manuscript. We carefully addressed your comments and believe that our paper has been considerably improved by this. Please find below our detailed answers to the points you have raised as well as an additional note on the changes applied to the manuscript during the course of revising the manuscript. The page numbers in our answers refer to the revised manuscript.

Response to reviewers

Reviewer #1: 

1. "When discussing about obesity paradox, it is important to acknowledge and account for the limitation of using BMI as the only measure of obesity. Previous studies have showed that waist circumference and waist-to-hip predicted cognitive performance better than BMI." 

We fully agree that BMI or any other weight-based measure alone is an insufficient measure. In fact, we tried to argue that it is necessary to complement it with a measure that allows for a differentiation between lean muscle mass and fat mass, where waist circumference or waist-to-hip ratio would be examples for measures of the latter (p.6). 

"I understand that the authors used grip strength, but I am not clear how this can address the limitation of the lack of waist circumference measure. More elaboration is necessary and this limitation should be noted in the discussion."

We used grip strength, which has been demonstrated to be correlated with lean muscle mass (see added references in the manuscript), particularly intra-personally as we employ it as a proxy for muscle mass. If it were a (close to) perfect measure, this would capture any possible effect related to lean muscle mass, while BMI would capture the residual effect related to fat mass. We tried to make this clearer in the revised manuscript (p.6) and added references that back up our interpretation (p.10).

However, we have to acknowledge that grip strength is probably not a perfect proxy. If waist circumference or waist-to-hip-ratio had been available in the data, we could have used one of them as complement instead. If we had done so, the residual effect of BMI would have to be interpreted as an effect of (changes in) lean muscle mass. Unfortunately, we are restricted to grip strength. On a conceptual level, it is still supposed to capture muscle mass, on the operational level, it may be suboptimal. We acknowledge and discuss this limitation in an additional paragraph added to the discussion (see 3rd paragraph in the discussion, p.25f).

2. "Another limitation that should be acknowledged and discussed is the cognitive measures employed in the current study. Only a short immediate and delayed recall tasks were used. The authors should discussed the validity of these tasks in measuring cognitive functions. How reliable is the tasks? Some previous studies have found that the link between obesity cognitive functions may depend on the tasks and the specific cognitive abilities that were measured. More discussion is necessary and possible limitations should be acknowledged appropriately."

Indeed, we only use a single combined measure for memory and recall to measure cognition and we agree that this fact deserves a more thorough discussion of implications and possible limitations. To address the raised concern, we therefore extended the paragraph on the used cognition measure and put it in the context of the different proclaimed domains of cognitive functioning (p.9). We also provide some arguments in favour of its use: It is based on an established and validated test that was found to be affected earlier and stronger by ageing than crystallised cognitive skills, such as numerical or verbal abilities. In addition, it has some desirable properties (from a statistical point of view), which warrant its use.

While we think the focus on memory/recall is justifiable, it is true that our results do not readily generalize to other domains of cognitive functioning. We acknowledge and discuss this limitation in an additional paragraph added to the discussion (see 4th paragraph in the discussion, p.26). We still believe in the importance of our contribution for this field of research – carefully scrutinizing possible mechanisms (including methodological pitfalls) that might contribute to the association of interest. We would highly appreciate if our study stimulates further research considering other domains of cognitive functioning.

Reviewer #2: 

Major revision:

"The selected participant average age is 66.94 years. As commented by the authors based on a review of the literature, Smith et al. have suggested that the mean age of the study sample may be a crucial factor with a positive weight-cognition association only emerging in samples with mean age above 72 years. However, the chosen age to perform the study is an age that is closer from the adulthood rather than elderly. Therefore, the study should be completed adding another group above 72 years, as suggested by the literature."

This is an excellent comment. We had indeed considered splitting our sample into young-old and older-old and had already looked at respective models in earlier versions of our manuscript. We opted against going deeper into this in order not to overburden the manuscript and given that we systematically address effect heterogeneity by age. On second thought, your suggestion offers a more straightforward link to the previous literature. However, since this literature is essentially cross-sectional, we felt it required drawing some more attention to the cross sectional models to first check whether we can replicate the finding of a (more) positive association in our data. For this reason, we decided to introduce basic control variables already at an earlier stage (i.e. into the POLS models) to be more comparable. 

We finally addressed your comment by splitting our sample into two age groups: one group, where respondents entered the panel at 65 years or above and another group for younger respondents. We chose this cut-point because it is close of the inflection point in the inversely u-shaped age profiles for both BMI and cognition in our data and subsample sizes are not overly unbalanced. Moreover, mean age in the older subsample is reasonably high with about 76 years on average (60 years for the younger). We then replicated our analyses separately for both sample splits and added two new paragraphs (p.21; p.22), where we summarize and discuss results in the manuscript. Full details, including test statistics for differences in coefficients across subsamples, are provided in the supplementary material.

In short, we found cross-sectional associations to be significantly higher (although not significantly positive) in the older sample. In the panel analyses, both weight gain and weight loss effects were more pronounced in the older sample, but differences were not significant; neither were the weight gain effects themselves. The recovery effect found for women was owed to older respondents, were it was more pronounced and did not show up among the younger (where the average effect of BMI increase was effectively zero in the first place). Overall, we think that the additional differentiation between young-old and older-old added some interesting further findings that strengthens our argumentation and improved the paper. Thank you very much.

Minor revision:

"Figure legends are missing."

We added a legend to Figure 1. 

Note on changes in the manuscript:

To adequately address Reviewer 2's comment, we decided to include control variables at an earlier stage then we did in the original manuscript. This affected subsequent model specifications and particularly the decomposition of the confounding factors, as age is now already included in the reference model. This slightly changed the results and their interpretation, but arguably in a more meaningful direction as it now considers omitted variable bias when age is controlled for (which it typically is). As a consequence, parts of the results section had to be adapted. Note that all analyses involving asymmetric fixed effects models with controls are unaffected by this change. Specifically, this refers to the paragraphs on recovery and effect heterogeneity that remained unchanged.

All changes applied to the manuscript are highlighted in red in the manuscript version with track changes. They are either a direct response to the reviewers or result from the model adjustments described. Also highlighted are changes correcting some copy editing errors we encountered (e.g. inconsistent use of tense or truncated numbers in the age range in table 1). 

Yours sincerely,

Judith Kronschnabl, also on behalf of my co-authors

---

## [Editor Report · Decision Letter 1]

23 Mar 2021

Bodyweight change and cognitive performance in the older population

PONE-D-20-38422R1

Dear Dr. Kronschnabl,

We’re pleased to inform you that your manuscript has been judged scientifically suitable for publication and will be formally accepted for publication once it meets all outstanding technical requirements.

Kind regards,

David Meyre

Academic Editor

PLOS ONE
---

## [Editor Report · Acceptance letter]

25 Mar 2021

PONE-D-20-38422R1 

Bodyweight change and cognitive performance in the older population 

Dear Dr. Kronschnabl:

I'm pleased to inform you that your manuscript has been deemed suitable for publication in PLOS ONE. Congratulations! Your manuscript is now with our production department. 

Kind regards, 

on behalf of

Dr. David Meyre 

Academic Editor

PLOS ONE